# Achieving model explainability for intrusion detection in VANETs with LIME

Fayaz Hassan[1], Jianguo Yu[1], Zafi Sherhan Syed[2], Nadeem Ahmed[3], Mana Saleh Al Reshan[4] and Asadullah Shaikh[4]

[1] Beijing Key Laboratory of Work Safety Intelligent Monitoring, School of Electronic Engineering, Beijing University of Posts and Telecommunications, Beijing, China
[2] Department of Telecommunication Engineering, Mehran University of Engineering and Technology Jamshoro, Jamshoro, Pakistan
[3] State Key Laboratory of Wireless Network Positioning and Communication Engineering Integration Research, School of Electronics Engineering, Beijing University of Posts and Telecommunications, Beijing, China
[4] Department of Information System, College of Computer Science and Information Systems, Najran University, Najran, Saudi Arabia



Corresponding author
Fayaz Hassan,
fayaz.hassan@bupt.edu.cn

## ABSTRACT

Vehicular *ad hoc* networks (VANETs) are intelligent transport subsystems; vehicles can communicate through a wireless medium in this system. There are many applications of VANETs such as traffic safety and preventing the accident of vehicles. Many attacks affect VANETs communication such as denial of service (DoS) and distributed denial of service (DDoS). In the past few years the number of DoS (denial of service) attacks are increasing, so network security and protection of the communication systems are challenging topics; intrusion detection systems need to be improved to identify these attacks effectively and efficiently. Many researchers are currently interested in enhancing the security of VANETs. Based on intrusion detection systems (IDS), machine learning (ML) techniques were employed to develop high-security capabilities. A massive dataset containing application layer network traffic is deployed for this purpose. Interpretability technique Local interpretable model-agnostic explanations (LIME) technique for better interpretation model functionality and accuracy. Experimental results demonstrate that utilizing a random forest (RF) classifier achieves 100% accuracy, demonstrating its capability to identify intrusion-based threats in a VANET setting. In addition, LIME is applied to the RF machine learning model to explain and interpret the classification, and the performance of machine learning models is evaluated in terms of accuracy, recall, and F1 score.

# INTRODUCTION

The integral part of the intelligent transport system (ITS) is smart autonomous vehicles (SAVS) for the future of automotive industries. SAVs are important in road safety, driving experience, and decision-making based on available information. To design the roadway information, the information was collected from different communication technologies and SAVs. The road infrastructure depends on the connection between vehicles and roadside units (RSUs), as shown in Fig. 1.

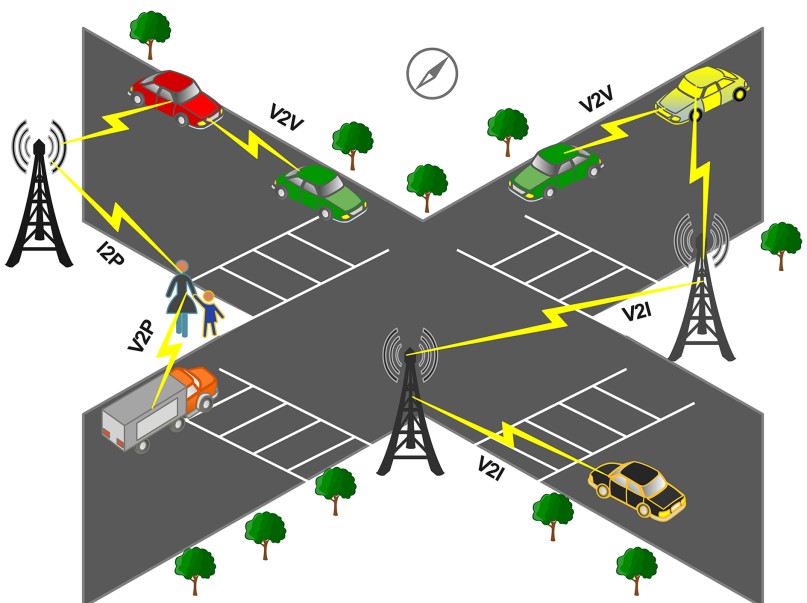

**Figure 1 General illustration of VANETs communication.**

   ITS utilizes RSUs to prevent accidents and improve driving performance (*Alsarhan et al., 2018*). Vehicles communicate through a wireless medium since the wireless medium is sophisticated for several attacks. The performance of vehicular *ad hoc* networks (VANETs) is degraded by these attacks, which can cause critical problems for drivers. Because of the different attacks in VANET, the protection of VANET traffic from tempering, deletion, and monitoring has become a major problem. It is one of the academic's and industry's top priorities to solve/tackle because the intruder can exploit this control over user data to disturb the network (*Cui et al., 2019*). The characteristics of VANET include high mobility, dynamic network topology and safety message. Due to high mobility, the network topology can change rapidly so the intrusion detection schemes for VANETs need to be able to adapt to these changes in real time. However, other wired and wireless network scenarios may have more stable network topologies that change less frequently, allowing for more static intrusion detection schemes. VANETs communicate through a wireless mediums, making them vulnerable to attacks. In other network scenarios, data may be less critical and have lower security requirements. In contrast, VANETs require strict security measures to ensure safety-critical information, such as collision avoidance and emergency response. Therefore, intrusion detection schemes used in VANETs need to be designed to protect against specific types of attacks, such as DOS attacks and their types, ping flood (sends a large number of Internet Control Message Protocol requests to a target device), UDP flood (targets the User Datagram Protocol by sending a large number of UDP packets to the target device), HTTP flood (targets web servers by sending a large number of HTTP requests), and syn flood (sends a large number of synchronised packets to the target but never completes the three-way handshake that establishes a connection). These attacks aim to increase unsolicited network traffic to

prevent authorized users from accessing resources. Vehicles have traditionally been manufactured without complete security requirements, making it difficult to protect them from intrusion. Using the premise that autonomous vehicles work without communication capability. When the connection grows, that restricts resources, creating a large attack surface and problem for the current vehicle. Encryption and access control are the traditional security countermeasures that are not useful for autonomous vehicles.

IDS is a reactive system have recently attracted increased attention. Intrusion detection in VANETs can be used to detect and prevent unauthorized network access or malicious network attacks (*Sepasgozar & Pierre, 2022*). This can be accomplished by employing specialized intrusion detection systems (IDS) that monitor network traffic and behaviour for anomalies and suspicious activity (*Ali et al., 2019*). The IDS can be integrated into the VANET infrastructure to monitor vehicle-to-vehicle communication. It can analyze network data and detect any unusual patterns or activities that may indicate an intrusion or attack. Some common methods of intrusion detection used in VANETs include:

1) Signature-based detection: This technique compares network traffic to a database of known attack signatures or patterns. Traffic resembling a recognized attack signature is marked suspicious, and the network administrator is notified.

2) Anomaly-based detection: This technique analyzes network traffic to spot any abnormal activity or communication patterns. Any traffic that deviates from the norm is identified as suspicious and alerted to the network administrator

3) Rule-based detection: This method involves defining a set of rules or criteria that network traffic must meet to be considered normal. Any traffic that does not meet these criteria is flagged as suspicious and alerted to the network administrator

Overall, using intrusion detection in VANETs can help ensure the security and reliability of the network (*Bangui, Ge & Buhnova, 2022*) and prevent unauthorized access or attacks that could compromise its functionality. We seek to answer the question, "What are the important features that contribute to the random forest classifier's decision-making process in classifying different types of VANET attacks, and how can these features be interpreted using the LIME toolkit?" Using local interpretable model-agnostic explanations (LIME) as an explainable AI tool to explain the decision-making process of the random forest classifier for identifying various attacks in VANETs can help fill the gap in knowledge regarding transparency in machine learning-based intrusion detection systems. This gap is crucial because it is challenging to rely on the decisions made by such systems without interpretability. This is especially true for sensitive applications like VANETs where the safety of human lives is involved. Thus, it is crucial to understand the reasoning behind the system's classification of a particular event as an attack so that human operators can take proper actions to avoid potential accidents. This research consists of the following contributions:

- The proposal of the DDoS/DoS attack detection framework, which performed efficiently with a higher accuracy rate and low computational cost,

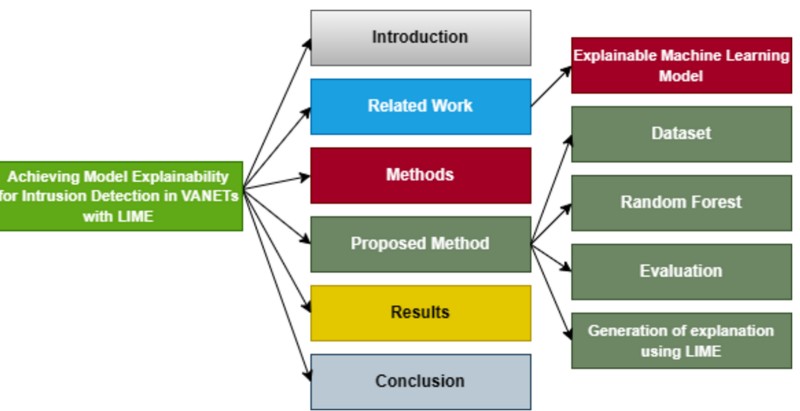

**Figure 2   Organization of this manuscript.**     

- ML algorithm applied to Application layer dataset to predict the network traffic by the vehicular communication
- To apply the random forest model to ML, evaluate the results in terms of accuracy, precision-recall, and F1 score
- Further, the authors implement the LIME for interpretation and a better understanding of the features of the ML model.

The organization of this article is illustrated in Fig. 2.

## RELATED WORK

In recent years, numerous academics have been interested in protecting VANETs (_Zhang et al., 2018_; _Engoulou et al., 2014_; _Zhang, Zulkernine & Haque, 2008_; _Huang et al., 2011_; _Liu et al., 2014_; _Li & Song, 2015_; _Chaubey, 2016_). There are numerous security vulnerabilities in VANET, which can render these VANET applications inoperable. Intrusion detection (IDS) is used to identify the attacks (internal and external) with high accuracy (_Kumar & Chilamkurti, 2014_; _Daeinabi, Rahbar & Khademzadeh, 2011_). Based on approaches that are previously applied, signature-based IDS and anomaly-based IDS are the common method of IDS (_Khraisat et al., 2020_).

Recently for DDoS attacks, several classification approaches have been developed. DDoS attacks at the protocol layer can be divided into two levels: the network layer and the application layer flooding attacks. Early detection and mitigation of impact are two of the key challenges encountered by DDoS attacks. However, it requires some important features that are unavailable from existing techniques (_Chang, 2002_). The Hypertext Transfer Protocol-based (HTTP) method for detecting HTTP flooding attacks using data sampling is provided in _Jazi et al. (2017)_. The study uses the CUMSUM (Cumulative Sum) algorithm to distinguish between malicious and normal communications. The number of requests made by the application layer and the number of zero-sized packets are used for traffic analysis. The result shows a detection rate between 80% to 86% per, with a sampling rate of 20%.

**Table 1 Summary of application and limitation for popular datasets.**

| Dataset | Reference | Year | Model | Scenario/application | Limitation/issues |
|---|---|---|---|---|---|
| DARPA | *MIT Lincoln Laboratory (1999)* | 1998 | MLP | Intrusion detection system data for background traffic. | Usage of artificially simulated |
| KDD Cup99 | *Ahmad et al. (2021)* | 1999 | MLP | Intrusion detection system computer networks | Numerous redundant record and data corruption resulted in biased testing outcomes. |
| CICDoS2017 DoS2019 | *Ahuja et al. (2021)* | 2021 | GRU, LSTM MLP, KNN, RF | DDoS attack detection for SDN | The suggested system for the transport layer and the application layer produces different outcomes for each since the application layer has a 95 attack detection rate Second, they employed intricate GRU models with large computational costs. |
| Application-Layer dataset | *Awan et al. (2021)* | 2021 | RF and MLP | Real-time DDoS attacks detection in real-time big data | They focus on both efficiency accuracy, but they only used two models, so we did not have enough data to determine the approach's impact. |
| Application-Layer dataset | *Rustam et al. (2022)* | 2022 | LR and GBM RF and ETC | DDoS attacks detection using the multi feature technique | They work on accuracy and reducing computational cost, using PCA and SVD for feature selection technique, they do not provide the feature with its weight/score |

In *Rustam et al. (2022)*, the work for detecting Dos/DDoS attacks was proposed. The author used a combination of two common techniques, principal component analysis (PCA) and singular value decomposition (SVD), to improve the model performance. Results indicate that this study achieved maximum accuracy, whereas the Application layer dataset was utilized for this research.

*Patil et al. (2022)* used an intrusion data detection system using a machine learning system, an SVM decision tree, and RF for classification. In this study, the authors used the CICIDS-2017 dataset. The LIME technique was used and achieved better accuracy. The classification compares decision trees, RF, SVM, and voting classifiers among them. Results showed that the accuracy of all of them was almost the same. The comparison of some available datasets is shown in Table 1.

## METHODS

This section will detail our proposed methodology for the intrusion method for VANETs. We shall start with a discussion on the explainability of machine learning models using LIME to provide context to our discussion before presenting our proposed method.

## Explainable machine learning models

There are various reasons why machine learning models should be interpretable. Explainable models, in general, promote improved comprehension and confidence in the model's judgments and forecasts. Explainable models can shed light on the variables the model employs to produce predictions, allowing for identifying and correcting any biases that may exist. Machine learning models are crucial for compliance, comprehension, trust, and spotting bias in decision-making procedures. There are numerous approaches to make machine learning models interpretable, such as employing rule-based models, which already have interpretability built in, or feature selection techniques, which can decrease the number of input variables and make the model more interpretable. Recent explainable AI techniques have gained popularity, including local interpretable model-agnostic explanations (LIME), which can explain the model's choices for specific predictions (*Hariharan et al., 2022*). These strategies can explain the model's predictions in a human-understandable style, such as the most important features that contributed to the prediction. While feature importance methods like random forest's built-in feature importance and feature selection can provide global insights into feature importance, they do not provide local interpretability. Local interpretability is provided by LIME, which explains how the classifier's judgement was reached for a single instance of data. This is especially valuable in intrusion detection systems when it is critical to understand why a specific event was labelled as an attack. Furthermore, unlike other feature importance approaches that are limited to specific types of models, LIME is model-agnostic, which means it may be applied to any form of model. LIME can handle complex models and data by approximating the model's decision boundary in the local region around a particular data instance, making it easier to interpret the feature importance.

## LIME

Local interpretable model-agnostic explanations (LIME) are a method for the interpretability of machine learning models. In LIME, a model's predictions are calculated for a given input sample. It provides a local explanation for individual predictions made by a machine learning model, allowing users to understand the factors that influenced the model's decision and identify any potential biases. In LIME, the input sample is then perturbed by randomly changing the values of some of its features, and predictions are re-calculated for each perturbed sample. The relationship between the perturbed features and the model's predictions is learned using a simple but interpretable model, such as linear regression or decision tree. The coefficients of the interpretable model are used to explain the factors that contributed to the model's prediction for the original input sample. Interpretability local surrogate model can be expressed mathematically in Eq. (1)

$$Explanation(x) = arg \min_{g \varepsilon G} L(f, g, \pi_x) + \Omega_g \tag{1}$$

Let us suppose x is the model g, which reduced the loss L (mean square error), which measures the explanation to the prediction with the original model f (RF model), prefers the important features for the model complexity $\Omega g$ is kept low. The possible explanation

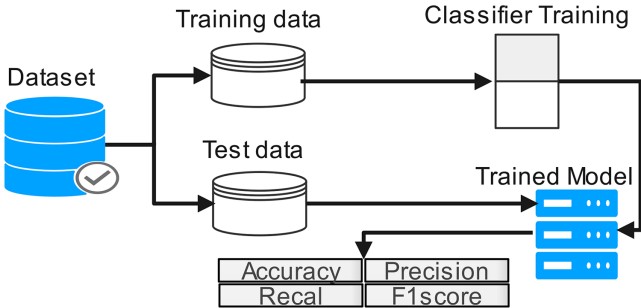

**Figure 3  Process flow of proposed methodology.**

is G (*e.g.*, for all possible models). The proximity measure pi(x) defines how large the neighbourhood around instance x is that we consider for the explanation.

## Proposed method

The proposed work diagram is shown in Fig. 3. Each step is described below

## Dataset

This study proposed the application layer DoS dataset, which can access from a well-known data set source, Kaggle (*Ghumman, 1999*). Application layer DoS attacks provide 78 attributes from 809,361 total entries. Some attributes are shown in Fig. 4 three classes are classified within the record by analyzing the network. Three classes are: (1) Benign, which can be described as lawful; (2) DoS Slowloris, which can be described as a DoS attack; and (3) DoS Hulk, which can be described as a DDoS attack.

Table 2 displays a sample of the dataset's records. In this study application layer, the DoS data set has been used. It can be accessed from Kaggle by accessing the network analysis. Three classes are classified among 809,361 records. The three classes are "Benign", "DoS Slowloris" and "DoS Hulk". To balance the data, we used 15,000 records from each class for our experiment. Table 2 shows a dataset of sample records (*Ghumman, 1999*). We chose the random forest classifier from the scikit-learn toolbox for our research since it is a well-established classification technique widely utilised in various domains and performed well on both low- and high-dimensional data. Our choice was also influenced by the ensemble aspect of the method, which allows it to reduce bias, variance, and over-fitting, a characteristic not present in many other algorithms.

## Random forest

Random forest is an ensemble technique supervised technique used for regression and classification. By using a supervised algorithm, random forests perform better results with a decision tree. To improve their accuracy, RF uses the bagging technique for the training process, and those models with high variance will be reduced. RF constitutes several decision trees from bootstrap samples to learn information about the fitting process. The major advantage of RF is that it is more effective than the other traditional classifiers because of their minimum classification error when dealing with a high data set. Other advantages are as follows: Get all features from the random forest and remove those

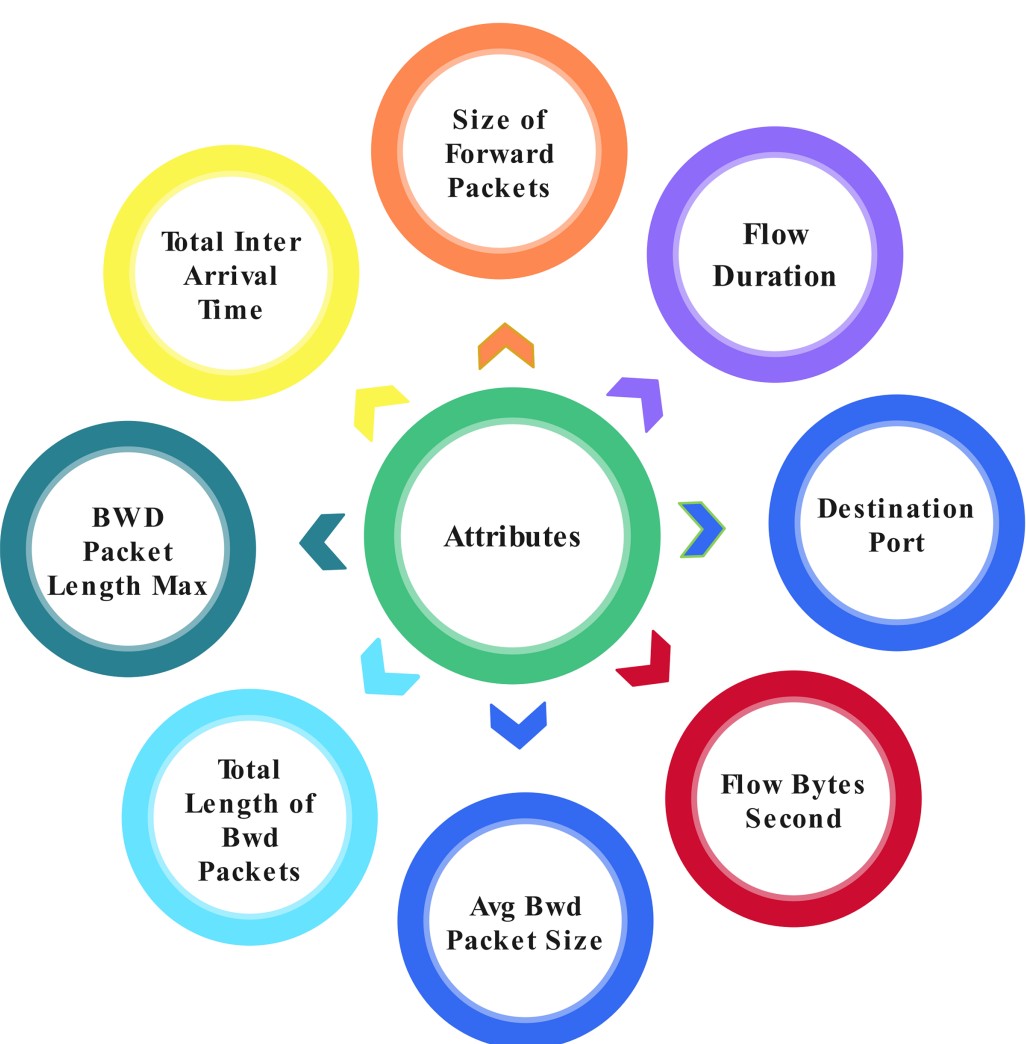

**Figure 4 Attributes in the dataset.**               

**Table 2 Summary of dataset partition and label distribution.**

| Partition | Benign | Dos Slowloris | DoS Hulk | Total |
|---|---|---|---|---|
| Train | 370,623 | 128,612 | 310,126 | 809,361 |
| Test | 159,295 | 55,180 | 132,394 | 346,869 |

features which are not important. Important features can be calculated from rank and RMS value. The random forest training model performance measurement is a collaborative model; it selects the random data tuples by creating a multiple data tree. The order to classify the features and eliminate the recursive feature is also an advantage of RF. Equation (2) expresses the RF algorithm. The sum of the feature's importance value on each tree is calculated and divided by the total number of trees:

$$RF_j = \frac{\sum j\varepsilon_{alltress} norm\ f_{ij}}{T} \tag{2}$$

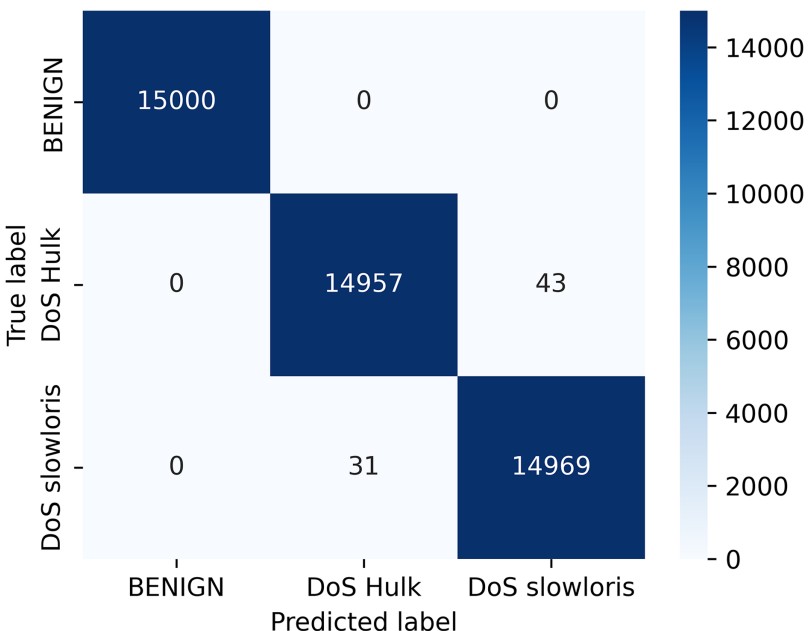

**Figure 5 Confusion matrix for classifier.**

whereas RF is the importance of feature is calculated from all trees in the random forest model, normfj is the normalized feature importance for i in tree j and T is the total number of trees.

## Evaluation

For model selection, it is usual practice to compare machine-learning algorithms based on their accuracy as a performance indicator. The performance metrics are accuracy, precision, and recall for comparing the machine learning algorithm. In many papers, accuracy is used for performance metrics, but it cannot evaluate the accurate result with one metric, most of the time, accuracy misleads the result as a performance metric, although incorrect prediction by classifier even that accuracy indicates with high score. Other metrics were added for the evaluation. The following are metrics for computation (*Patil et al., 2022*)

$$Precision = \frac{TP}{TP + FP} \tag{3}$$

$$Recall = \frac{TP}{TP + FN} \tag{4}$$

$$Accuracy = \frac{TP + TN}{TP + FP + TP + FN} \tag{5}$$

where TP represents the number of true positives, TN is the number of true negatives, FP is the number of false positives, and FN is the number of false negatives.

Figure 5 depict the confusion matrix for the random forest classifier, which is the relationship between actual and predicted values. The result shows the calculation of model accuracy and explains the relationship between the matrices, true positive (TP) with

false positive (FP) and true negative (TN) with false negative (FN) accuracy of our model defined in Fig. 5.

## Generation of explanations using LIME

First, we need to train the model before prediction. By using the network traffic of the application layer data set, training the model, and acquiring several features of interpretability. After splitting the train/est data, a random classifier fit it on the training set and obtained more than 100% accuracy; for better accuracy model interpretation technique, LIME was applied to the model. Figure 6 shows the flow diagram with the LIME technique. The LIME model was added to the machine learning model to improve lime. The LIME can explain many machine learning prediction models with accuracy. The 'change of feature' value used the convert the data sample into the transforming individual feature score into the contribution of prediction. Each data sample can be contextually explained by score. For instance, the model of interpretable LIME is frequently trained using minute perturbations (such as adding random noise, eliminating specific words, and hiding portions of an image) for linear regression or decision trees (*Ribeiro, Singh & Guestrin, 2016*; *Adadi & Berrada, 2018*). Organizations that trust their predictions can use machine learning models more broadly. However, the question of building trust in machine learning models persists. After applying the local interpretable model (LIME), the model diagram explains the issue in detail with black-box classifiers. LIME is a method for understanding a black-box machine learning model by perturbing the input data samples, and also it can be observing how the predictions vary. Any machine-learning black box model can be applied using LIME. The following are the main steps:

- Initially tabular explainer initialized the data which can be used to train (or the statistics of the training data if there are no training data), feature information, and, if classification is possible, details about the class name.
- In the explain method, accept the reference for the instances required for the explanation, including the number of features in the explanation and prediction of the trained model.

  The LIME explanation is described in three sections as follows

- In the left section, confidence in prediction has been shown.
- In the center section, the most important features are shown, and 10 are the important features. The Min-Seg-Size-Forward, Flow-Packets-Sec, Destination-Port, Flow-Bytes-Sec, Avg-Bwd-Segment-Size, Total-Length-of-Bwd-Packets, Bwd-Packet-Length-Max, Fin-Flag-Count, Bwd-Iat-Total and Bwd-Psh-Flags, features are selected. These features differentiate into two colors. A blue color represents class 1, whereas class 0 represents the grey color. The feature's contribution can be seen in the horizontal lines.
- In the rightmost section, the important features with their score value are differentiated with different colors shown
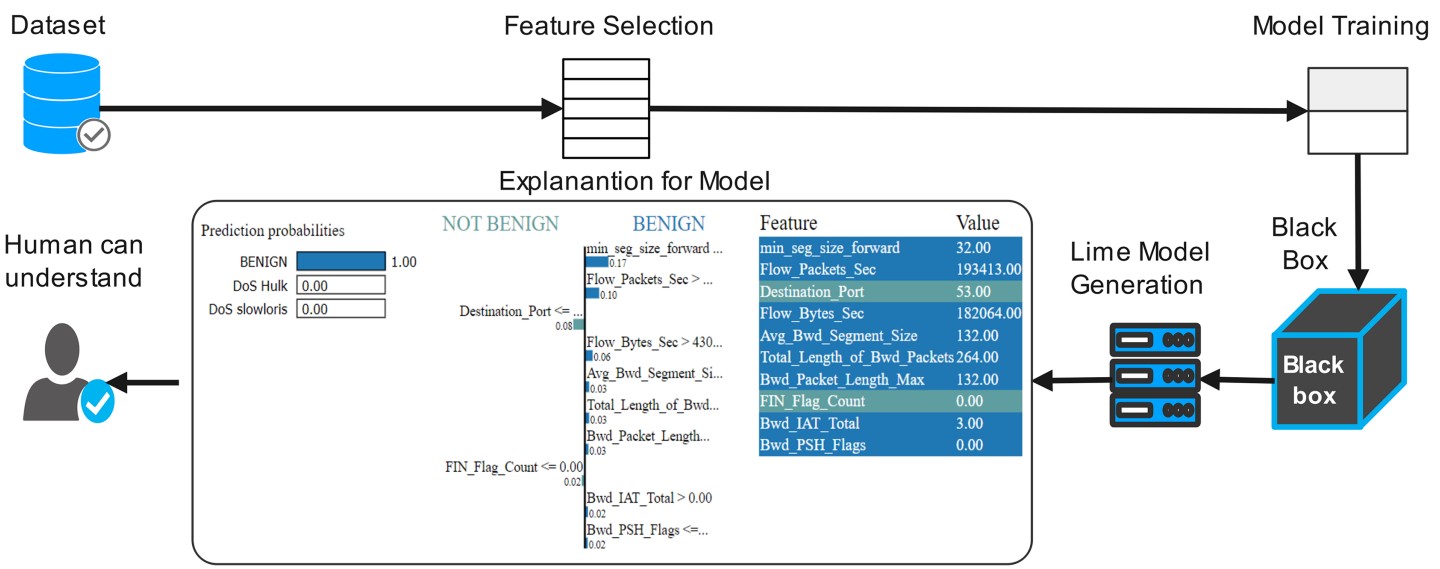

**Figure 6 Proposed flow diagram with explainable (LIME) technique.**

The LIME technique is effectively used for the classification, regression, and supervised models (*Gunning et al., 2019*; *Ribeiro, Singh & Guestrin, 2016*). The decision framework is predicated on the premise all complex models at the local level are linear. The LIME model simulates the complicated model's behaviour at one location to explain the simple model's predictions at another location. LIME is compatible with tabular, textual, and image data types.

The LIME explanation algorithm is defined as follows: If an explanation is necessary, there must be n occurrences of disruption without a modest change in value. Using these fabricated data, LIME builds a local linear model, including the perturbed observation.

- The outcomes of data perturbations are predicted.
- Determine the distance between perturbation and the actual observation.
- Compare the score by using their distance.
- To calculate features that best describe the predictions derived from the altered data.
- A basic model is fitted to the perturbed data by using the selected features.
- The coefficients (weights for features) of the simple model characterize the observations.

The model building is described by the pseudocode as follows

```
        Import  necessary  libraries
Load  dataset
Preprocess  dataset
        Split  dataset  into  features  (X)  and  labels  (y)
        Perform  feature  scaling
Create  train-test  partitions
Train  Random  Forest  classifier
```

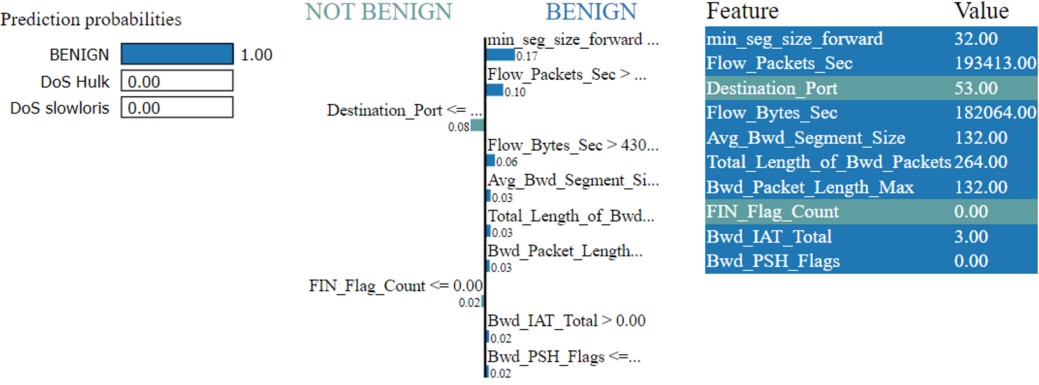

**Figure 7** Explanation of the LIME observations for benign.

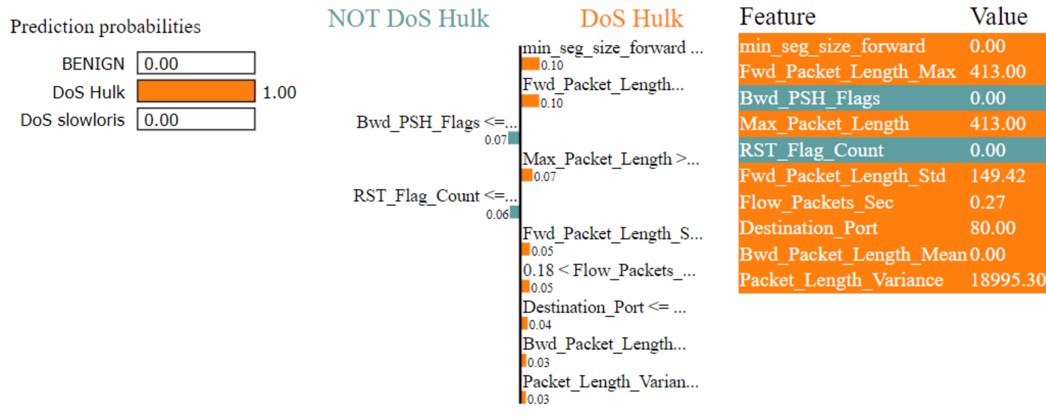

**Figure 8** Explanation of the LIME observations for DoS Hulk.

```
          Initialize  Random  Forest  classifier  with
appropriate  hyperparameters
        Evaluate  the  model
              Predict  labels  for  the  test  data  (test_X)
              Calculate  performance  metrics
      Apply  LIME  for  model  interpretability
              Initialize  LIME  explainer
              Create  explanations  for  a  sample  of  test
instances
              For  each  test  instance  in  the  sample
              Generate  LIME  explanation
                Display  feature  importance  and  explanations
      Report  results  (e.g.,  performance metrics, LIME  explanations)
```

Figures 7–9 explain the LIME observations for three classes using random forest. LIME explains why the probability was first allocated. The prediction is computed by comparing the probability values to the target variable. However, the probability ranged from −1 to

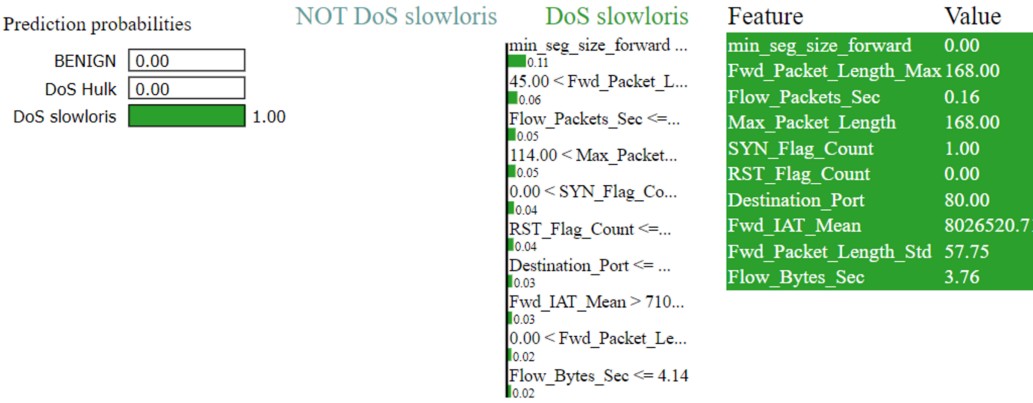

**Figure 9 Explanation of the LIME observations for DoS Slowloris.**

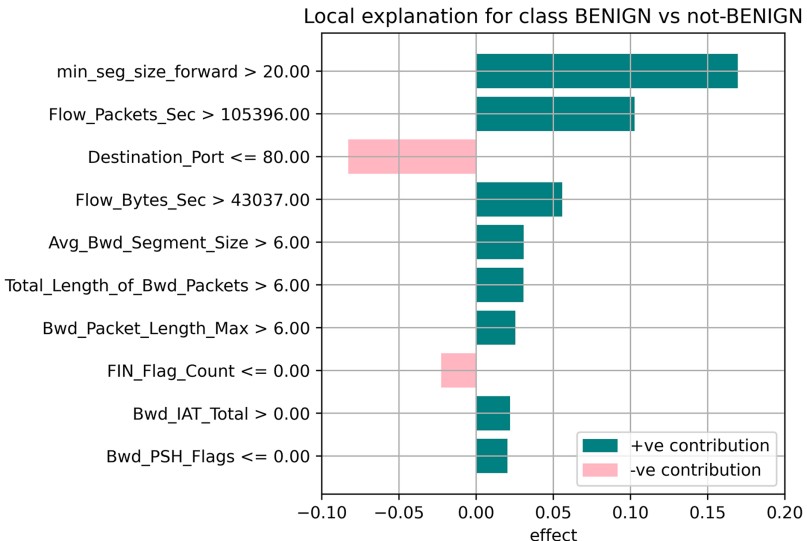

**Figure 10 Local explanation for benign.**

1.0. For example, the value of min-seg-size-forward is 32 and was assigned a weight of 0.17 for the BENIGN, a weight of 0.105 for DoS HULK, and 0.11 for DoS SLOWLORIS. Different color coding is assigned to each feature to indicate their contribution to the prediction in the feature value table, such as BENIGN (blue), DoS HULK (orange), and DoS SLOWLORIS (green).

## RESULTS

### Local explanation for random forest model

The random forest method used the estimate the relevance of features in the regression problem, as shown in Figs. 10–12. The data analysis library Scikit-learn provides a random forest classifier and random forest regression class that can make the importance of features in three classes which are Benign, DoS, and DoS Hulk. Feature importance property can be queried to each feature of relative weight after applying the fitted model.

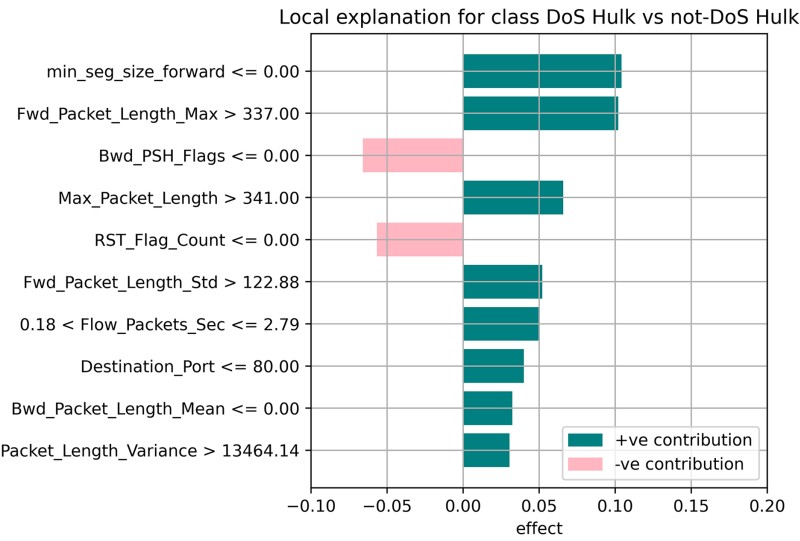

Figure 11 Local explanation for Dos Hulk.   

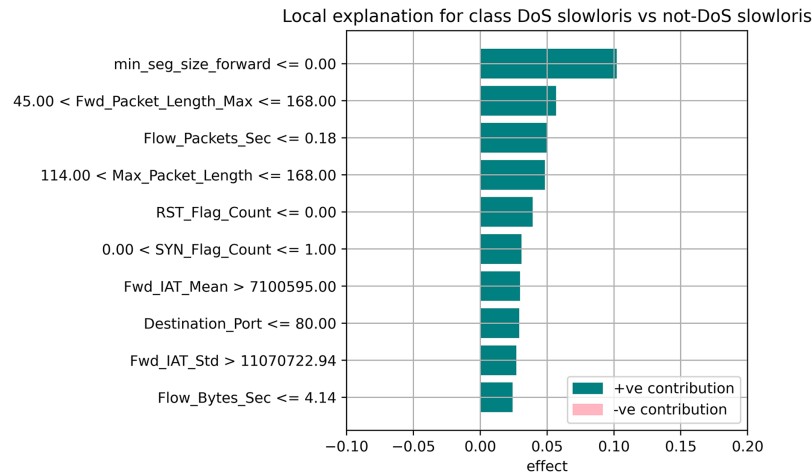

Figure 12 Local explanation for Dos Slowloris.   

Random forests were used to estimate the relevance of features in a classification problem with 15,000 samples and 10 features.

The relevance of split points is computed using decision tree approaches such as classification and regression trees (CARTs), which use common metrics such as Gini and entropy to determine split points. Similarly, ensembles of decision trees can be applied to algorithms such as a random forest. The feature importance property can be accessed after a model has been fitted to retrieve the relative importance scores for each input feature. Figures 10–12 represent the calculated value for each feature. Each factor contributing to the prediction is depicted in the bar graph on the right. The right sidebar represents the positive influence of the feature on the target, while the left sidebar represents the negative impact. For example, min-seg-size-forward, flow-packets-sec, and flow-bytes-sec

contribute more, as shown in Fig. 10. The positive coefficients for the predictors indicate that raising their values increases the model's feature scores.

## DISCUSSION

The features identified by the LIME toolkit can be used to answer the research question on the interpretability of machine learning-based intrusion detection systems in VANETs. More specifically, these features can provide insights into the decision-making process of the random forest classifier and help understand the characteristics of different types of attacks in VANETs. For instance, the feature "min-seg-size-forward" refers to the minimum segment size observed in the traffic flow's forward direction, which can indicate certain types of attacks. The feature "flow-packets-sec" refers to the rate of packets flowing through a flow, which can indicate the intensity of an attack. Similarly, the feature "destination-port" refers to the port number a packet is destined for, which can help identify the type of service or application targeted by an attack. By analyzing the importance of these features, we can gain insights into how the random forest classifier can differentiate between different types of attacks and what features are most important in making these decisions. This information can improve the classifier's accuracy by identifying the most informative features for intrusion detection in VANETs. Moreover, it can provide insights into the characteristics of different types of attacks, which can be used to develop more effective countermeasures against these attacks.

## CONCLUSIONS

Our research addresses a critical knowledge gap in the field of VANETs. We presented a novel machine learning-based intrusion detection solution for VANETs. We investigated a random forest classifier-based machine learning framework for detecting DDoS/DoS attacks at the application layer. Our approach was further enhanced by using the LIME model-agnostic method for describing predictions clearly and understandably. By providing prediction explanations, LIME proved to be highly beneficial for selecting representative models, improving unreliable models, and obtaining insights into predictions. Our evaluation criteria included accuracy, precision, recall, and F1-score, and our ML algorithms demonstrated high performance, achieving an accuracy of 99.84% for the IDS prediction. We computed the accuracy based on the confusion matrix, where all 15,000 samples from the Benign class were identified correctly, whereas 14,957 and 14,969 samples from the DoS Hulk and DoS slowloris classes were identified out of a total of 15,000 samples for each of the two classes. We determined the accuracy as (15,000 + 14,957 + 14,969)/45,000 = 0.99835. We converted the result to percentage and rounded it up to two decimal places to get 99.84%.

The LIME explanation graphs also illustrated the prediction performance of the random forest methods. Our findings highlight the potential of our approach for improving the accuracy and interpretability of intrusion detection in VANETs, and we suggest further research in this area to build upon our results.

### Funding

This research was funded by the Beijing Key Laboratory of work Safety Intelligent Monitoring (Beijing University of Post and Telecommunication), National Natural Science Foundation of China, Grant Number 62127802. The funders had no role in study design, data collection and analysis, decision to publish, or preparation of the manuscript.

### Grant Disclosures

The following grant information was disclosed by the authors:
Beijing Key Laboratory of Work Safety Intelligent Monitoring (Beijing University of Post and Telecommunication).
National Natural Science Foundation of China: 62127802.

### Competing Interests

The authors declare that they have no competing interests.

### Author Contributions

- Fayaz Hassan conceived and designed the experiments, analyzed the data, authored or reviewed drafts of the article, and approved the final draft.
- Jianguo Yu conceived and designed the experiments, analyzed the data, authored or reviewed drafts of the article, and approved the final draft.
- Zafi Sherhan Syed conceived and designed the experiments, analyzed the data, authored or reviewed drafts of the article, and approved the final draft.
- Nadeem Ahmed performed the experiments, performed the computation work, prepared figures and/or tables, and approved the final draft.
- Mana Saleh Al Reshan performed the experiments, performed the computation work, prepared figures and/or tables, and approved the final draft.
- Asadullah Shaikh performed the experiments, performed the computation work, prepared figures and/or tables, and approved the final draft.

### Data Availability

The dataset is available at Kaggle: https://www.kaggle.com/code/hamzasamiullah/ml-analysis-application-layer-dos-attack-dataset/notebook.
The data and code are available at Zenodo: Fayaz Hassan. (2023). application layer data set/Lime interpretability code. https://doi.org/10.5281/zenodo.7789616.

### Supplemental Information

Supplemental information for this article can be found online at http://dx.doi.org/10.7717/peerj-cs.1440#supplemental-information.

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
