# Peer review of "Achieving model explainability for intrusion detection in VANETs with LIME"

_PeerJ Computer Science, doi:10.7717/peerj-cs.1440_

## Round 0.1 · original submission · Major Revisions

· Academic Editor

Major Revisions

The paper needs some modifications, many of which are mentioned by the reviewers.

The proposed work needs to be well described by a block diagram, flowcharts, and/or pseudocodes.

A comparison with existing work needs to be highlighted.

·

Basic reporting

The Paper sounds good for the ITS field.
The Paper is well organized and needs more pefition in writing the equations.
Fgure 4, "Attributes in the dataset" has some typing mistakes as [Totoal???interarrival time ]
Many words written in small letters whereas it is Titles (first letter mst be Capital)
Symbols are not matched in the text and equationsas "OMEGA (g)" (as an example)
Figure 5. Confusion matrix resolution is very bad and can not be read??????

Experimental design

The submission defines the research question.
The knowledge gap being investigated is not clear enough
The gap is not supported by means of flow chart or pesudocode.
The study try to fill the gap .
The supplemental files need more descriptive metadata identifiers to be useful to future readers.
Although the presented results are compelling, the data analysis should be improved in the following ways: mean, varience, expected errors, relation to data set, and type of DOS attacks.
why the Random forests were used to estimate the relevance of features in a classification problem with 15000 samples and 10 features? (Based on what and what about the staility of the outcomes for such small number of samples???)

Validity of the findings

Data on Table 1" Comparison of some available datasets" has no citation and or refrence ???
Data on Table 2. "records avaialble in datasets" has no citation and or refrence ???
The metrics for computation are neither "defined via citation and or refrence ???" nor comapred to previously published work(s).

Additional comments

In the Conclusion part, authors said that -line 178- "The outcomes demonstrated an increase in
accuracy to 99.84% for the IDS prediction, and the LIME explanation graphs illustrated the prediction
performance of the random forest methods."
My question is, on war bases or cacluation could the autjors said 99.84% accuracy??????
NO VALIDATION or Comparisons had been presented.
Convolution of the proposed algorithm is not represented or mentioned.

·

Basic reporting

1. The author needs to redesign the pictures in the paper, especially the part of V2X communication in Figure 1. Some details could be added to show the characteristics of VANET. It is also worth noting that some parts of pictures 5 and 6 are too vague, and the proportion of figure 2, 3 and 4 also needs to be modified.

Experimental design

2. In addition to the LIME mentioned in this article, there are still other interpretable methods. Please explain the reasons for choosing LIME.
3. Many scholars have made in-depth research on intrusion detection methods based on random forests. Please compare the methods proposed in this paper with others.

Validity of the findings

4. The author needs to further introduce the application layer traffic characteristics of VANETs and explain the difference between VANETs' intrusion detection scheme and other network scenarios.

---

## Round 0.2 · Minor Revisions

· Academic Editor

Minor Revisions

Please consider all your replies to the previous version's comments to be included in the submitted version, particularly regarding the accuracy calculation.

·

Basic reporting

Now the paper has become eligible enough to be accepted

Experimental design

Now the paper has become eligible enough to be accepted

Validity of the findings

Now the paper has become eligible enough to be accepted

Additional comments

Now the paper has become eligible enough to be accepted

But I think the accuracy calculation presented in the rebuttal letter should be included in the final version of the manuscript

---

## Round 0.3 · accepted · Accept

· Academic Editor

Accept

You have sufficiently addressed the important reviewers' comments.